# Peer J

# Beyond birth-weight: early growth and adolescent blood pressure in a Peruvian population

Robie Sterling[1,2], William Checkley[1,2,3,4], Robert H. Gilman[1,2,3],
Lilia Cabrera[2], Charles R. Sterling[5], Caryn Bern[6] and
J. Jaime Miranda[1,7]

[1] CRONICAS Center of Excellence in Chronic Diseases, Universidad Peruana Cayetano Heredia, Lima, Peru
[2] Asociación Benéfica PRISMA (A.B. PRISMA), Lima, Peru
[3] Program in Global Disease Epidemiology and Control, Department of International Health, Bloomberg School of Public Health, The Johns Hopkins University, Baltimore, MD, USA
[4] Division of Pulmonary and Critical Care, School of Medicine, The Johns Hopkins University, Baltimore, MD, USA
[5] Department of Veterinary Science and Microbiology, University of Arizona, Tucson, AZ, USA
[6] Global Health Sciences, and Department of Epidemiology and Biostatistics, School of Medicine, University of California San Francisco, San Francisco, CA, USA
[7] Department of Medicine, School of Medicine, Universidad Peruana Cayetano Heredia, Lima, Peru

Corresponding author
J. Jaime Miranda,
Jaime.Miranda@upch.pe

## ABSTRACT

**Background.** Longitudinal investigations into the origins of adult essential hypertension have found elevated blood pressure in children to accurately track into adulthood, however the direct causes of essential hypertension in adolescence and adulthood remains unclear.

**Methods.** We revisited 152 Peruvian adolescents from a birth cohort tracked from 0 to 30 months of age, and evaluated growth via monthly anthropometric measurements between 1995 and 1998, and obtained anthropometric and blood pressure measurements 11–14 years later. We used multivariable regression models to study the effects of infantile and childhood growth trends on blood pressure and central obesity in early adolescence.

**Results.** In regression models adjusted for interim changes in weight and height, each 0.1 SD increase in weight for length from 0 to 5 months of age, and 1 SD increase from 6 to 30 months of age, was associated with decreased adolescent systolic blood pressure by 1.3 mm Hg (95% CI −2.4 to −0.1) and 2.5 mm Hg (95% CI −4.9 to 0.0), and decreased waist circumference by 0.6 (95% CI −1.1 to 0.0) and 1.2 cm (95% CI −2.3 to −0.1), respectively. Growth in infancy and early childhood was not significantly associated with adolescent waist-to-hip ratio.

**Conclusions.** Rapid compensatory growth in early life has been posited to increase the risk of long-term cardiovascular morbidities such that nutritional interventions may do more harm than good. However, we found increased weight growth during infancy and early childhood to be associated with decreased systolic blood pressure and central adiposity in adolescence.

## INTRODUCTION

The prevalence of hypertension has increased dramatically in the past two decades among adults in developing nations (*Kearney et al., 2004*) and globally among adolescents and children (*McNiece et al., 2007*; *Stranges & Cappuccio, 2007*; *Jackson, Thalange & Cole, 2007*). Currently, high blood pressure is the leading risk factor for death globally and is the cause of an estimated 57 million disability adjusted life years (*WHO, 2009*).

Longitudinal investigations into the origins of adult essential hypertension have found elevated blood pressure in children to accurately track into adulthood (*Bao et al., 1995*), however the direct causes of essential hypertension in adolescence and adulthood remains unclear. The developmental origins of disease theory posits that characteristic growth patterns during the prenatal period (often proxied as birth weight or length), infancy (birth to two years), and childhood (2–9 years) predict subsequent risk for elevated adult blood pressure and essential hypertension (*Eriksson et al., 2000*; *Adair & Dahly, 2005*; *Barker, 2006*).

To date, the association of these three developmental periods on adolescent and adult blood pressure has varied in both reproducibility and magnitude. Low birth weight has been well established to predict higher adolescent and adult blood pressure (*Eriksson et al., 2000*; *Adair & Dahly, 2005*; *Adair & Cole, 2003*; *Law et al., 2002*; *Adair et al., 2009*; *Huxley, Shiell & Law, 2000*; *Horta et al., 2003*). However, the degree of association has differed across populations and some investigations were unable to replicate these findings (*Hemachandra et al., 2006*; *Huxley, Neil & Collins, 2002*; *Menezes et al., 2007*). Evidence regarding the influence of weight gain in infancy on adolescent and adult blood pressure remains inconclusive, with studies showing positive (*Singhal et al., 2007*; *Kark, Tynelius & Rasmussen, 2009*; *Jones et al., 2012*) and negative (*Adair & Cole, 2003*) associations, or no association (*Law et al., 2002*). Likewise, increased rates of childhood weight gain have also been linked to later blood pressure (*Eriksson et al., 2000*; *Adair & Cole, 2003*; *Law et al., 2002*; *Huxley, Shiell & Law, 2000*; *Horta et al., 2003*). Yet the clinical importance of this association remains difficult to determine, as increased weight gain is also highly predictive of obesity (*Adair et al., 2009*), a confounding factor, as BMI has been found to positively associate with blood pressure (*Ford, Nonnemaker & Wirth, 2008*).

We sought to address the following research question: how does growth rate in infancy and early childhood, independently, affect blood pressure and central adiposity in adolescence in a middle-income country? To answer this, we took advantage of a longitudinal cohort of Peruvian infants tracked in infancy via monthly growth measurements and revisited them at adolescence.
## MATERIALS AND METHODS

The study setting was Pampas de San Juan de Miraflores, a peri-urban shanty-town (pueblo joven) located 25 km south of Lima, Peru. Initially settled in the early 1990s, with temporary structures lacking water or sewage lines, the community has undergone many economic and social developments over the last two decades. By 2008, over 75% of homes were constructed from brick or cement with in-home water and sewage. Improved transportation within this community has changed patterns of exercise as "moto-taxis" have decreased the need to walk from main byways to the residential regions. The diet, while maintaining many typical aspects, now is more similar to an urban setting, an energy dense diet. The region continues its transition from a communicable to a chronic disease society, with current rates of hypertension ranging from 12.6% and 14.4 in private urban clinics (*Schargrodsky et al., 2008*), to 19.5 and 11.4 in residents of Pampas de San Juan de Miraflores, for men and women respectively (*Davies et al., 2008*). Further details of this community are described elsewhere (*Gilman et al., 1993*; *Berkman et al., 2002*; *Checkley et al., 2003*).

### Exposures measured in infancy

The enrollment of children into the original birth cohort study was conducted from February 1995 to December 1998 (*Checkley et al., 2003*; *Checkley et al., 2002*; *Checkley et al., 2004*). Beginning in 1995, participants were enrolled at no more than 6 months of age and were tracked via daily diarrhea surveillance and monthly measurements of height and weight. Length was measured to the nearest 0.1 cm with a locally made wooden platform and sliding footboard and weight to the nearest 0.1 kg with Salter scales (Salter Housewares LTD, Tonbridge, England).

The anthropometric data gathered in the original 1995 study were reanalyzed in 2008 to calculate length for age (LAZ) and weight for length (WLZ) *z*-scores using the 2006 World Health Organization (WHO) child growth standards developed for children from age 0 to 5 years (*Bloem, 2007*). LAZ is used in place of height for age (HAZ) *z*-score in younger children due to the measurement being made while the child is lying on their back. Based on standard international classifications, for baseline variables, stunting was calculated as LAZ $< -2$ standard deviations (SD), underweight and overweight as a WLZ $< -2$ and $> 2$ SD, respectively. Diarrhea was evaluated by daily home visits. A diarrhea day was defined as a day on which the mother reported the child had diarrhea and the child had passed at least three liquid or semi-liquid stools. A diarrhea episode began on the first diarrhea day and ended on the last diarrhea day followed by two consecutive days without diarrhea. Information on breastfeeding was collected and has been evaluated and published elsewhere (*Checkley et al., 2002*). Based on our previous analyses, we have found that little heterogeneity in patterns of breastfeeding: mixed breastfeeding is prolonged in all children and that exclusive breastfeeding ends early in life ($<$3–4 months) for virtually all participants. Therefore in the past we have found it difficult to identify breastfeeding subgroups that are not collinear with age and thus breastfeeding information were not considered in this analysis.

## Follow-up assessment and outcomes

Between April 2008 and July 2009, we conducted a follow-up assessment in those eligible for this study (*Sterling et al., 2012*). Participants were re-contacted and written informed consent was obtained from their parents if they were enrolled in the initial study at no later than three months of age and completed at least 12 months of longitudinal follow-up. This round consisted of a parental socioeconomic survey, anthropometric and blood pressure measurements. The socioeconomic survey explored housing condition, food access, maternal and paternal education and living accommodations. Height was measured to the nearest 0.1 cm with a locally constructed wooden stadiometer, waist circumference and hip circumference were measured to the nearest 0.1 cm using a tape measure. Blood pressure and heart rate measurements were made using an automated monitor (Omron HEM 742). The same trained individual made all measurements, in triplicate. The mean of the three measurements were used in analysis, with the exception of blood pressure, where the mean of the latter two values was used. The data were then used to calculate body mass index (BMI) (kg/height$^2$) and waist-to-hip ratio (WHR).

BMI for age (BAZ) *z*-score at follow-up was calculated using the 2007 WHO anthropometric reference for children and adolescents (*de Onis et al., 2006*), standardized for individuals aged 5–19. Socioeconomic survey data was used to adjust for socioeconomic status (SES). Maternal education and number of people per room in 2008 were used as proxies of socioeconomic deprivation, long-term and current SES respectively. We defined deprived SES as participants whose mothers did not receive education beyond primary school and homes with a concentration of greater than 3 people per room of a residence.

The outcomes of interest were systolic and diastolic blood pressure, waist circumference and WHR.

## Biostatistical analysis

Data were analyzed in two stages, as described in greater detail elsewhere (*Sterling et al., 2012*). In the first stage, we estimated the longitudinal prevalence rate of diarrhea as the number of days with diarrhea in the first year of life over total child-days followed multiplied by 100 for each age at which measurements were made.

We then estimated the slopes of change for LAZ and WLZ in infancy as a function of age and sex using a random-effects model (*Laird & Ware, 1982*) while adjusting for longitudinal prevalence of diarrhea, at each monthly data point in infancy, to account for growth variations due to infection. In exploratory analyses, we found the change in LAZ per month of age to be relatively linear, while WLZ growth consistently had two slopes with an inflexion at ∼6 months of age (*Sterling et al., 2012*). Therefore, we summarized growth in LAZ using a single slope and WLZ in infancy using two slopes, one from 0 to 5 months of age, and another from 6 to 30 months, identified as infancy and early childhood. The intercept values (predicted for month zero) were used as proxies of birth LAZ and WLZ, and interpreted as a measurement of prenatal growth (*Sterling et al., 2012*).

In the second stage, we used a sequence of multiple linear regression models to assess the influence of growth in infancy and early childhood on the outcomes of interest.

These analyses were performed independently for LAZ and WLZ in infancy. We used the following regression models: Model 1 adjusted for sex, age at the time of follow-up, and socioeconomic deprivation; and Model 2 adjusted for the variables in Model 1 as well as adolescent BAZ and current height. Adjustment for adolescent BAZ addressed any growth that occurred in between the end of serial measurement in infancy and the follow-up visit. Additionally, we adjusted for current height to minimize overestimating the influence of our exposures of interest, LAZ and WLZ at birth and the rate of change in each during infancy, due to its strong correlation to blood pressure.

Additionally, we stratified growth rates in tertiles and compared the mean blood pressure values for each tertile. All analyses were performed using STATA version 9.2 (STATA Corp., College Station, TX).

### Ethics

The original 1995 study was approved by the Institutional Review Boards of A.B. PRISMA (Lima, Peru) and the Johns Hopkins School of Public Health (Baltimore, MD, USA). Institutional Review Boards of A.B. PRISMA and Universidad Peruana Cayetano Heredia, both in Lima, Peru, approved the follow-up study.

## RESULTS

### Cohort characteristics

In the 2008–2009 follow-up visit, we completed measurement on 75% (147 out of 196) of the eligible participants (Fig. 1). We found no differences in early change in height ($p = 0.387$) or weight ($p = 0.834$), maternal height ($p = 0.172$), the percentage of households without sewage connection ($p = 0.077$) or in the type of house flooring ($p = 0.860$) between the 147 participants who completed follow-up and the remaining 49 (25%) participants of the original cohort that met inclusion criteria but did not participate in the follow-up assessments.

Table 1 shows summarized characteristics of the participants. Average age at enrollment for the group was 13.4 days and duration of diarrheal surveillance was 24.7 months. During the initial study in 1995, 46.9% of participants were stunted at some point in infancy. Rates of underweight and overweight were 5.5 and 57.2%. The rate of stunting at follow-up was 13.6%, while 1.4% and 10.2% were underweight vs. overweight, respectively.

### Relationship between early growth and blood pressure in adolescence

We first examined the association between growth rates early in development and adolescent blood pressure. Table 2 shows the results of our multivariable regression of growth during development on adolescent blood pressure. Neither LAZ at birth nor change of LAZ was significantly associated with blood pressures measures in adolescence (Table 2). In Model 2, WLZ at birth showed a borderline significant association with systolic blood pressure, but none with diastolic blood pressure. The rate of change measures for WLZ were significantly associated with systolic but not diastolic blood

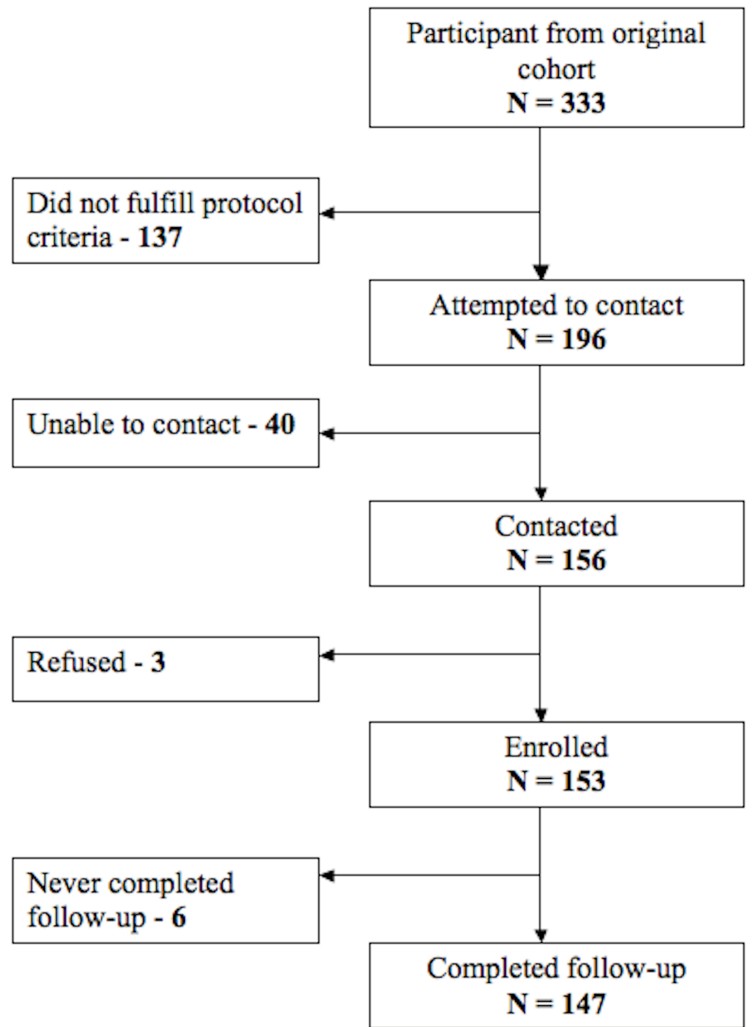

**Figure 1 Participant follow-up flow chart.** Description of number of participants in each stage of the study and its follow-up.

pressure at follow-up when adjusted for adolescent BAZ and height, i.e., increased growth was associated with decreased adolescent systolic blood pressure. For each 0.1 SD increase in WLZ from 0 to 5 months of age and 1 SD increase from 6 to 30 months of age, adolescent systolic blood pressure decreased by the order of 1.3 mm Hg and 2.5 mm Hg, respectively. When analyzed growth rates by tertiles, there was no evidence of an association with blood pressure (data not shown).

## Relationship between early growth and central adiposity in adolescence

We next examined the influence of variable growth rates in infancy on adolescent central adiposity. LAZ at birth and rate of change in LAZ in infancy were strongly associated with adolescent waist circumference in multivariate regression unadjusted for adolescent body

Table 1 **Participant characteristics at baseline and follow-up.**

| | Total | Male | Female | *p*-value |
|---|---|---|---|---|
| Sample size | 147 | 81 | 66 | |
| **Baseline 1995–1998** | | | | |
| Age of entry into study (days), mean ± SD | 13.4 ± 14.4 | 14.6 ± 15.2 | 11.9 ± 13.4 | 0.01 |
| Follow-up time (months), mean ± SD | 24.7 ± 5.7 | 23.7 ± 6.4 | 26.1 ± 4.5 | 0.033 |
| Birth weight for length *z*-score (WLZ), mean ± SD | 0.79 ± 1.0 | 0.58 ± 1.0 | 1.1 ± 1.0 | 0.066 |
| Birth length for age *z*-score (LAZ), mean ± SD | −0.14 ± 0.9 | −0.21 ± 0.9 | −0.05 ± 1.0 | 0.313 |
| Stunted from 0 to 30 months (LAZ < −2 SD), % | 46.9 ± 0.5 | 51.9 ± 0.5 | 40.9 ± 0.5 | 0.189 |
| Overweight from 0 to 30 months (WLZ > 2 SD), % | 57.2 ± 0.5 | 53.7 ± 0.5 | 61.5 ± 0.5 | 0.349 |
| Diarrhea prevalence, % | 2.39 ± 2.57 | 2.41 ± 2.40 | 2.36 ± 2.79 | 0.33 |
| Diarrhea episodes per month, mean ± SD | 0.27 ± 0.22 | 0.27 ± 0.21 | 0.26 ± 0.23 | 0.622 |
| **Follow-up 2008–2009** | | | | |
| Age (years), mean ± SD | 13.4 ± 0.76 | 13.3 ± 0.70 | 13.4 ± 0.70 | 0.379 |
| Height (cm), mean ± SD | 150.4 ± 7.4 | 151.1 ± 8.2 | 149.5 ± 6.2 | 0.188 |
| Stunted (LAZ < −2 SD), % | 13.6 ± 0.3 | 8.6 ± 0.3 | 19.7 ± 0.4 | 0.052 |
| Overweight (BAZ > 2 SD), % | 10.2 ± 0.3 | 12.3 ± 0.3 | 7.6 ± 0.3 | 0.345 |
| Systolic blood pressure (mm Hg), mean ± SD | 103.3 ± 8.8 | 104.0 ± 8.1 | 102.4 ± 9.6 | 0.249 |
| Diastolic blood pressure (mm Hg), mean ± SD | 63.0 ± 7.2 | 63.0 ± 6.9 | 63.0 ± 7.6 | 0.987 |
| Waist circumference (cm), mean ± SD | 71.1 ± 7.9 | 70.8 ± 7.9 | 71.4 ± 8.0 | 0.679 |
| Waist to hip ratio, mean ± SD | 0.84 ± 0.05 | 0.86 ± 0.05 | 0.82 ± 0.05 | <0.01 |
| **Socioeconomic traits 2008** | | | | |
| Educationally-deprived mother, % (*n*) | 68.5% (102) | 71.1% (59) | 65.2% (43) | 0.405 |
| Child repeated a grade, % (*n*) | 26.2% (39) | 21.7% (18) | 31.8% (21) | 0.164 |
| People per room ≥3, % (*n*) | 6.0% (9) | 3.6% (3) | 9.1% (6) | 0.166 |
| In home sewage, % (*n*) | 87.3% (130) | 88.0% (73) | 86.4% (57) | 0.842 |

parameters (see Model 1 in Table 3). Each 1 SD increase in LAZ at birth was associated with a 3 cm increase in waist circumference, and each 1 SD per month increase in LAZ from 0 to 30 months of age was associated with a 1.8 cm increase in waist circumference. However, when adjusted for adolescent BAZ and height, these associations became attenuated and non-significant. Neither LAZ at birth nor rate of change in LAZ in infancy was associated with WHR.

Similarly, WLZ at birth was positively associated with both waist circumference and WHR prior to adjustment, but these estimates attenuated once adjusted for BAZ and height. The rate of change in WLZ in early and late infancy was also positively associated with waist circumference and WHR prior to adjustment for adolescent body size. However, after adjustment, these relationships with adolescent adiposity became inversed and significant, with more rapid growth in infancy associating with a smaller waist circumference in adolescence.

**Table 2 Adolescent systolic and diastolic blood pressure by growth parameters during early life[a].**

| | Systolic blood pressure (mm Hg) β-coefficient (95% CI) | | Diastolic blood pressure (mm Hg) β-coefficient (95% CI) | |
|---|---|---|---|---|
| | Model 1[b] | Model 2[c] | Model 1 | Model 2 |
| LAZ at birth | 1.59 (−0.2; 3.4) | −0.97 (−3.2; 1.3) | 0.26 (−1.2; 1.7) | −0.35 (−2.3; 1.6) |
| Rate of change LAZ 0–30 months[d] | 0.80 (−0.6; 2.2) | −0.94 (−2.6; 0.7) | −0.26 (−1.4; 0.9) | −0.71 (−2.1; 0.7) |
| WLZ at birth | 0.60 (−2.0; 3.2) | −2.44 (−5.1; 0.2) | 0.54 (−1.6; 2.7) | −0.85 (−3.2; 1.5) |
| Rate of change WLZ 0–5 months[e] | −0.23 (−1.4; 0.9) | **−1.25 (−2.4; −0.1)** | 0.05 (−0.9; 1.0) | −0.38 (−1.4; 0.6) |
| Rate of change WLZ 6–30 months | −0.77 (−3.3; 1.8) | **−2.45 (−4.9; −0.0)** | −0.02 (−2.1; 2.1) | −0.78 (−3.0; 1.4) |

**Notes.**

[a] Birth weight and length were estimated via slope calculations.

[b] Model 1: adjusted for sex, age at the time of follow-up, and socioeconomic deprivation.

[c] Model 2: adjusted for the variables in Model 1 as well as adolescent BAZ and height.

[d] Rate of change in LAZ based on 1 SD/month change.

[e] Rate of change in WLZ based on 0.1 SD/month and 1 SD/month change for 0–5 and 6–30 months, respectively.

**Table 3 Adolescent measures of central adiposity by growth parameters in early life[a].**

| | Waist circumference (cm) β-coefficient (95% CI) | | WHR (*100) β-coefficient (95% CI) | |
|---|---|---|---|---|
| | Model 1[b] | Model 2[c] | Model 1 | Model 2 |
| LAZ at birth | **2.97 (1.5; 4.5)** | 0.67 (−0.4; 1.7) | 0.12 (−0.8; 1.0) | 0.22 (−0.7; 1.1) |
| Rate of change LAZ 0–30 months[d] | **1.78 (0.6; 3.0)** | 0.1 (−0.64; 0.85) | −0.11 (−0.8; 0.6) | −0.14 (−0.8; 0.5) |
| WLZ at birth | **4.41 (2.3; 6.5)** | −1.19 (−2.4; 0.0) | **1.65 (0.4; 2.9)** | −0.42 (−1.5; 0.7) |
| Rate of change WLZ 0–5 months[e] | **1.27 (0.3; 2.2)** | **−0.57 (−1.1; −0.0)** | 0.34 (−0.2; 0.9) | −0.31 (−0.8; 0.2) |
| Rate of change WLZ 6–30 months | 1.96 (−0.1; 4.1) | **−1.19 (−2.3; −0.1)** | 0.56 (−0.7; 1.8) | −0.61 (−1.6; 0.4) |

**Notes.**

[a] Birth weight and length were estimated via slope calculations.

[b] Model 1: adjusted for sex, age at the time of follow-up, and socioeconomic deprivation.

[c] Model 2: adjusted for the variables in Model 1 as well as adolescent BAZ and height.

[d] Rate of change in LAZ based on 1 SD/month change.

[e] Rate of change in WLZ based on 0.1 SD/month and 1 SD/month change for 0–5 and 6–30 months, respectively.

## DISCUSSION

Our longitudinal study aimed to assess the effects of early growth on adolescent blood pressure and central adiposity by taking advantage of serial measurement of anthropometric indicators, with data accruing monthly for a period of up to 30 months, and thus overcoming limitations posed by single ascertainment of birth weight or modelling techniques that approximate growth patterns. We found increased rates of weight gain in infancy were strongly associated with decreased systolic blood pressure and waist circumference but not diastolic blood pressure, 10 years later, at adolescence. Weight gain in early and late infancy, respectively, predicted decreases in systolic blood pressure of 1.3 and 2.5 mm Hg, and decreases in waist circumference of 0.6 and 1.2 cm. This is consistent with findings observed in a population of Filipino adolescent males (*Adair & Cole, 2003*)

where increased weight gain from 0 to 2 years of age resulted in decreased odds of high adult blood pressure in males. Notably, no association was found between weight gain and diastolic blood pressure, which recent evidence suggests may be a stronger correlate to later mortality than systolic blood pressure (*Sundstrom et al., 2011*). The magnitude of the decrease in systolic blood pressure observed in this study, via improved weight gain during infancy, is equivalent to that achieved via resource-intensive community-based lifestyle interventions in Pakistan (*Jafar et al., 2010*).

The public health implications of these findings are substantial given that blood pressure in teens has been found to track into adulthood. The magnitude of reductions in blood pressure observed in our cohort could be linked to significant gains in terms of cardiovascular disease outcomes. In a meta-analysis of 61 prospective observational studies of blood pressure and vascular disease in adults, for each 2 mm Hg decrease in systolic blood pressure, to a minimum of 115 mm Hg, stroke mortality and cardiovascular mortality decreased by 10% and 7%, respectively (*Lewington et al., 2002*).

Our findings contrast with recent reports from Belarus (*Tilling et al., 2011*) and the United Kingdom (*Jones et al., 2012*) in which growth from birth was associated with increases in blood pressure in childhood and at early adolescence, respectively. Their findings from settings with advanced child health outcomes, e.g., low rates of stunting and child mortality, may indicate that, in healthy infants, increased rates of early growth pose increased cardiovascular risk as there is less need for compensatory growth following growth delays and stunting, both commonly encountered in the developing country settings. On the contrary, additional evidence from other similar resource-poor settings aligns with our observations. Albeit weak, hospital admissions due to diarrhea during the first year of life have been linked to high blood pressure in childhood (*Miranda et al., 2009*), and undernutrition has been linked to increased cardiovascular risk (*Sawaya et al., 2003*).

Much as growth during development has been correlated with later blood pressure, it has also been found to predict an increased risk of overweight and obesity. We found increased weight at birth and rate of weight gain in early and late infancy to be associated with decreased adolescent waist circumference, a measure of central adiposity and cardiovascular risk (*Huxley et al., 2010*). Birth length and growth in length in infancy were not associated with the degree of central adiposity in adolescence. These findings indicate that in this population, of generally undernourished children, rapid weight gain during infancy may help decrease the risk for central adiposity in adolescence, possibly by preventing compensatory weight gain later in childhood when the effects on cardiovascular risk are greater. This is consistent with findings in Brazil, in which rapid growth during infancy in males was associated with increased adolescent lean mass, while rapid growth in childhood was associated with fat mass (*Wells et al., 2005*). This appears to indicate that increased growth rates early in development have a protective effect on adult cardiovascular risk factors such as stunting (*Sterling et al., 2012*), obesity (*Wells et al., 2005*) and hypertension (*Adair & Cole, 2003*).

Our study has a number of strengths. While other studies looking at the developmental origins of disease relied, at most, on yearly anthropometric measurements, frequent serial

measurement of our cohort afforded us enough power to find effects of early growth on systolic blood pressure. The use of short interval serial measurements early in life and a relatively novel methodology offered an accurate characterization of growth early in development. This characterization of growth benefited uniquely from the longitudinal diarrheal data, which was incorporated in our modeling to account for growth variations due to infection. Thus, our work provides further advancements on the relationships of early life exposures to adolescent outcomes in countries with a high burden of childhood chronic under nutrition.

We also considered various potential limitations to our analysis. First, we did not directly measure birth weight or length, however there was a mean age of entry into the study of 14 days, with many individuals entering in the first days of life. Second, our estimations were adjusted for maternal education and overcrowding as proxies of long-term and current SES respectively. We considered that since the time of the initial study, maternal education was unlikely to have changed and thus a good proxy of SES at baseline (*Howe et al., 2012*). To account for socioeconomic variation over time, since the community was settled, we used number of people per room, a measure of SES at follow-up. Despite this approach, we did not, however, address socioeconomic change over time and this may have introduced some bias. As described earlier, most of the community setting progressed to meet basic needs by the time of follow-up, yet in general this community remains a resource-deprived area compared to Lima (*Instituto Nacional de Estadística e Informática, 2007*). Third, we did not include breastfeeding as an explanatory variable in our model; however, since both type and duration of breastfeeding are strongly associated with growth patterns in early childhood (*Baker et al., 2004*; *Singhal & Lanigan, 2007*), breastfeeding is likely in the causal pathway of the relationship growth trajectories in early childhood and blood pressure in later life. Hence, we opted not to include breastfeeding in our analysis. Moreover, a large, long-term follow-up of 13,879 mother–infant pairs who were randomized to receive a breastfeeding promotion intervention vs. usual care did not prevent overweight or obesity, nor did it affect IGF-I levels at 11.5 years of age (*Martin et al., 2013*). We also did not formally assess degree of pubertal development, which may have altered our outcomes in regards to growth as well as blood pressure. Finally, our sample size did not enable sex stratification that could disentangle potential differences as with Filipino adolescents (*Adair & Cole, 2003*).

As has been described by Lucas and colleagues (*Lucas, Fewtrell & Cole, 1999*), the inclusion of current size in the regression model is an adjustment for growth between the time of exposure and time of follow-up. Consistent with findings in other populations, adjustment for current size caused the beta coefficient for WLZ at birth to shift sign from positive to negative (*Lucas, Fewtrell & Cole, 1999*). This positive to negative shift indicates that, holding childhood growth constant, increased weight growth in the first 30 months of life may have a protective effect on adolescent blood pressure levels.

In low- and middle-income countries undergoing rapid economic development there is an increased risk of concurrent stunting and overweight, due to infectious disease associated growth retardation early in development (*Victora et al., 2008*) followed by

overnutrition in adolescence and adulthood (*Popkin, 2001*). Rapid compensatory growth in underweight children has been posited to increase the risk of long-term cardiovascular morbidities such that nutritional intervention may do more harm than good (*Singhal et al., 2007*). However, we found increased weight growth during infancy and early childhood to predict decreases in adolescent systolic blood pressure and central adiposity. The impact of such findings at the population level, as indicated by the magnitude of the change in systolic blood pressure observed in this study, aligns with significant decreases in vascular mortality in the future (*Lewington et al., 2002*). Thus, in addition to the known short term (*Victora et al., 2001*) and long term (*Martorell et al., 2010a*; *Martorell et al., 2010b*) benefits of increased growth in underweight children, increased weight growth early in development may also decrease rates of risk factors for cardiovascular disease and its associated health outcomes.

## NOVELTY AND SIGNIFICANCE

### What is new?
- Taking advantage of a daily diarrheal surveillance study, this study benefits from monthly anthropometric measurements in infancy.
- This study moves beyond single ascertainment of birth weight or the growth curve modeling techniques to address the relationship between growth in infancy and blood pressure in adolescence.

### What is relevant?
- The developmental origins of disease theory posits that characteristic growth patterns during the prenatal period, infancy and childhood predict subsequent risk for elevated adult blood pressure and essential hypertension.
- To date, the association of these three developmental periods on adolescent and adult blood pressure has varied in both reproducibility and magnitude.

### Summary
- We found increased weight growth during infancy and early childhood to predict decreases in adolescent systolic blood pressure and central adiposity.

## ACKNOWLEDGEMENTS

Our special gratitude to the staff and the team of fieldworkers that contributed to different parts of this study. Most importantly, our sincere gratitude is extended to the people that agreed to take part in the study.

### Funding

The original study was funded by an International Centers for Tropical Disease Research (ICTDR) grant from the National Institutes of Allergy and Infectious Diseases awarded to the Johns Hopkins Bloomberg School of Public Health (U01-A135894). J. Jaime Miranda,

Robert H. Gilman and William Checkley were supported by National Heart, Lung, and Blood Institute's Global Health Initiative under the contract Global Health Activities in Developing Countries to Combat Non-Communicable Chronic Diseases (Project Number 268200900033C-1-0-1). William Checkley was further supported by a Clinician Scientist Award from the Johns Hopkins University and a K99/R00 Pathway to Independence Award (K99HL096955) from the National Heart, Lung and Blood Institute, National Institutes of Health. Robie Sterling was supported by a pre-doctoral T35 Training Grant (T35AI065385) of the National Institutes of Health. The funders had no role in study design, data collection and analysis, decision to publish, or preparation of the manuscript.

### Grant Disclosures

The following grant information was disclosed by the authors:
International Centers for Tropical Disease Research (ICTDR), National Institutes of Allergy and Infectious Diseases: U01-A135894.
National Heart, Lung, and Blood Institute's Global Health Initiative under contract Global Health Activities in Developing Countries to Combat Non-Communicable Chronic Diseases: Project Number 268200900033C-1-0-1.
Clinician Scientist Award, Johns Hopkins University.
K99/R00 Pathway to Independence Award, National Heart, Lung and Blood Institute: K99HL096955.
Pre-doctoral T35 Training Grant, National Institutes of Health: T35AI065385.

### Competing Interests

JJM is an Academic Editor for PeerJ. Robie Sterling, William Checkley, Lilia Cabrera and Robert H. Gilman work with the Asociación Benéfica PRISMA.

### Author Contributions

- Robie Sterling conceived and designed the experiments, performed the experiments, analyzed the data, wrote the paper, prepared figures and/or tables, reviewed drafts of the paper.
- William Checkley conceived and designed the experiments, analyzed the data, wrote the paper, reviewed drafts of the paper.
- Robert H. Gilman conceived and designed the experiments, contributed reagents/materials/analysis tools, reviewed drafts of the paper, investigator of the original diarrhea study in children.
- Lilia Cabrera performed the experiments, contributed reagents/materials/analysis tools, organized, provided supervision to fieldwork activities to follow-up adolescent study and took part in original diarrhea study.
- Charles R. Sterling and Caryn Bern contributed reagents/materials/analysis tools, investigator of the original diarrhea study in children.
- J. Jaime Miranda conceived and designed the experiments, performed the experiments, wrote the paper, reviewed drafts of the paper.

## Human Ethics

The following information was supplied relating to ethical approvals (i.e., approving body and any reference numbers):

The original 1995 study was approved by the Institutional Review Boards of A.B. PRISMA (Lima, Peru): CE1116.08 (approval letter); CE0704.10 (renewal letter) and the Universidad Peruana Cayetano Heredia: SIDISI 54673.

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
