# Peer review of "Beyond birth-weight: early growth and adolescent blood pressure in a Peruvian population"

_PeerJ, doi:10.7717/peerj.381_

## Round 0.1 · original submission · Minor Revisions

This paper presents findings from a follow-up study of children initially included in a cohort study relating to diarrhoea and growth, in Lima Peru. These children were then re-contacted approximately 10 years after they were last seen, for one assessment of their weight, height, waist circumference and some socio-demographic questions. The study has been peer-reviewed elsewhere, and conforms to the PeerJ requirements.

Specific comments to address:

Title: seems to be not quite correct – it is unclear what ‘Improved ascertainment’ would be? The study notes as its strength the repeat growth measurements upto 30 months of age, but that in itself does not constitute an improvement.

Abstract:

Given that an abstract is often read in isolation, it is important it can sort of stand-alone. Thus be clear about the ages of the children at enrollment in the original study, whether this was an open cohort or not, age at last visit in the original study. Further, childhood growth needs to be more clearly defined. Both in the abstract and throughout the manuscript, replace ‘predictors/predicting’ with ‘associated with’ as it is not correct to infer causality from a cohort study such as this one.

Main text

Methods: line 109: Were all children enrolled in early 1995, all shortly after they were born and then all followed up for 30 months, or was this an open cohort with enrolment throughout the period 1995-1998 – please clarify.

Methods, line 113 – please clarify that no information was collected on breastfeeding

Methods, line 122 – please clarify the assessment of diarrhoeal episodes – was this as reported by the mother at each of the monthly visits, did the mother keep a diary which was then assessed at monthly visits or what? Was the absence of diarrhoea equally reliably reported?

Methods, line 125: Please clarify the consent procedure – who was approached, the parent or the child? Given the ages of the children presumably the parent had to give consent and the child assent?

Methods, line 129: who reported in the socio-economic survey: the parent or the child?

Methods, line 148: Unclear what ‘prevalence of diarrhoea days’ is? Should this be ‘number’?

Methods, line 150 – please clarify how the adjustment for longitudinal prevalence of diarrhoea worked – is this just allowing for the number of days with diarrhoea or number of episodes of chronic or acute diarrhoea in the first year or first two years of life in the regression analyses of LAZ and WLZ?

Methods, line 154-155 – please provide relevant references re the methodological approach used.

Results, line 181: it would be helpful to see more details on the n=137 children who were not contacted as they did not fulfil inclusion criteria for the follow-up study, to assess possible selection bias.

Results, line 188: please clarify what is meant by ‘ stunted at some point in infancy’. Clearly it would be more relevant to have a variable that is ‘stunted or not’ at a given age of say one or two years, or when last seen in the initial study or use it as a time-varying variable given you have monthly assessments? In terms of understanding the relationship and the complexity it will be important to be clear about the dynamics of the early growth, before evaluating the association with growth measurements ten years later.

Results, Table 1 – it is usually more informative to provide median and range, rather than mean and SD for variables such as age at visit, BP etc, as the mean can be skewed by a few outlyers.

Results, line 196 – text says there was a borderline statistically significant association between WLZ at birth and systolic blood pressure – but that is not entirely obvious from Table 2, although it could be the case in Model 2 only? Please clarify.

Results, line 201: it is also important to stress that there was no significant association between WLZ and diastolic blood pressure. And then to come back to this in the discussion.

Results, line 210 – it would be helpful to more clearly discuss the ‘attenuation’ of the size of the associations after allowing for adolescent BAZ and height, and consider in the discussion what that ‘attenuation’ implies.

Discussion, line 222 – note in this context modelling is with double ‘l’

Discussion, line 264: discuss why there was only a statistically significant association with systolic, but not diastolic, blood pressure. And clarify that this was approx. 10 years later, in early puberty. The discussion should also note clearly that there were no data on whether the children assessed in adolescence had reached puberty or not (or what stage they were at) – this is relevant for both growth and blood pressure assessment.

Discussion, line 281: the lack of data on breastfeeding is noted as a limitation, but it would be helpful to indicate what the ever breastfed rates are in this setting, and what the average duration of breastfeeding is. This may be important and of interest, as the fast growth seen in early infancy in the UK etc may more likely than in Peru be associated with non-breastfeeding.

Table 1 – clarify who repeated a grade, the child or the mother

Table 2 – clarify infancy (infancy is upto one year of age usually)

External reviews were received for this submission. These reviews were used by the Editor when they made their decision, and can be downloaded below.

---

## Round 0.2 · accepted · Accept

No further comments. Thanks for the changes and edits made.

External reviews were received for this submission. These reviews were used by the Editor when they made their decision, and can be downloaded below.